# Sol–Gel Synthesis of Translucent and Persistent Luminescent SiO_2_@ SrAl_2_O_4_ Eu, Dy, B Materials

**DOI:** 10.3390/ma16124416

**Published:** 2023-06-15

**Authors:** Madara Leimane, Katrina Krizmane, Ivita Bite, Jurgis Grube, Virginija Vitola

**Affiliations:** Institute of Solid State Physics, University of Latvia, Kengaraga str. 8, LV-1063 Riga, Latvia; madara.leimane@cfi.lu.lv (M.L.); katrina.krizmane@cfi.lu.lv (K.K.); ivita.bite@cfi.lu.lv (I.B.); jurgis.grube@cfi.lu.lv (J.G.)

**Keywords:** persistent luminescence, glass, sol–gel synthesis

## Abstract

This publication offers an economically promising method of persistent luminescent silicate glass synthesis that does not involve high temperatures or ready-made (separately synthesized) PeL particles. In this study, we demonstrate the formation of SrAl_2_O_4_ doped with Eu, Dy, and B in a SiO_2_ glass structure using the one-pot low-temperature sol–gel synthesis method. By varying the synthesis conditions, we can use water-soluble precursors (e.g., nitrates) and a dilute aqueous solution of rare-earth (RE) nitrates as starting materials for SrAl_2_O_4_ synthesis, which can be formed during the sol–gel process at relatively low sintering temperatures (600 °C). As a result, translucent, persistently luminescent glass is obtained. The glass shows the typical Eu^2+^ luminescence and the characteristic afterglow. The afterglow duration is about 20 s. It is concluded that the slow drying procedure (2 weeks) is optimal for these samples to sufficiently get rid of the excess water (mainlyOH groups) and solvent molecules that can influence the strontium aluminate luminescence properties and have a pernicious effect on the afterglow. It can also be concluded that boron is playing a crucial role in the formation of trapping centers needed for PeL processes in the PeL silicate glass.

## 1. Introduction

Persistent luminescence (PeL), or afterglow, is a form of emission that continues for minutes or hours after the removal of the excitation source. The persistent luminescence properties of Eu and Dy-doped strontium aluminate luminophores include a long fading time, unsurpassed intensity, and good chemical stability [1,2]. The already wide field of possible applications of PeL materials is currently being supplemented by ongoing research work on embedding the material in different mediums, such as cloth, polymers, and glass systems [3,4,5,6,7,8,9].

SrAl_2_O_4_:Eu, Dy is the commonly used material employed in commercial applications for persistent luminescence. It is a spinel-type material with a bandgap of 6.4 eV [10,11]. Spinel-type materials have many important applications as optical materials or materials for nuclear/fusion technology, such as diagnostic/detector components and materials for shielding, etc. [12,13]. When compared to alternative materials, it surpasses them in terms of luminescence lifetime, boasting an impressive duration exceeding 20 h, which is unmatched by other materials. This material demonstrates chemical stability, good resistance to photodegradation, and an exceptionally vivid afterglow. As a result, it has garnered considerable attention through extensive research efforts [1,2]. Strontium aluminates exhibit different crystallographic forms that share the same broadband emission from the Eu luminescent transition; however, the afterglow time and the emission maximum vary for the different crystallographic forms (i.e., Sr_4_Al_14_O_25_, SrAl_4_O_7,_ and others) [14,15].

Translucent PeL glass enables the excitation source to be placed on any side of the material and offers the mechanical durability of glass combined with the optical properties of conventional PeL materials. A glass matrix is also beneficial due to the possibility to excite the whole material volume, not only the surface layer, as it is for non-translucent materials [16,17].

There have been efforts to synthesize PeL glass before—the “frozen sorbet” method was the first technique developed to prepare persistent luminescent crystals in glass [18,19]. However, this method requires very high temperatures (at least 1500 °C) and a fast heating rate, as well as separately synthesized PeL particles. Recently, it was shown that phosphate and borosilicate glasses with persistent luminescence properties can be prepared via melting of batches containing Sr_4_Al_14_O_25_:Eu^2+^ and Dy^3+^ microparticles conventionally in air [20,21]. This method also used previously synthesized particles that were subjected to heat treatment above 1200 °C.

Here we report an economically promising method of persistent luminescent silicate glass synthesis that does not involve high temperatures or ready-made (separately synthesized) PeL particles—the so-called “one-pot synthesis”. To our knowledge, this is the first report on this kind of synthesis for PeL silicate glass. Additionally, the influence of boron addition on the photoluminescence of PeL silicate glass is investigated.

## 2. Materials and Methods

### 2.1. Materials

Tetraethyl orthosilicate (TEOS, assay ≥ 99.0%, Sigma Aldrich, Taufkirchen, Germany), strontium nitrate (Sr(NO_3_)_2_ purity 98%, Sigma Aldrich, Germany), aluminum nitrate nonahydrate (Al(NO_3_)_3_·9H_2_O, purity 99.6%, VWR Prolabo Chemicals, Leuven, Belgium, LOT: 17E294134), europium oxide (Eu_2_O_3_, purity 99.99%, Alfa Aesar, Kandel, Germany, LOT: U04C022), dysprosium oxide (Dy_2_O_3_, purity 99.9%, Alfa Aesar, Germany LOT: B04 × 020), and boric acid (H_3_BO_3_) were used as the starting materials for the preparation of SrAl_2_O_4_:Eu, Dy, and B/SiO_2_ composite materials.

Ammonium acetate (AcONH_4_, assay 100.0%, Lach-Ner, Neratovice, Czech Republic, LOT: PP2017/12682) was used as a pH stabilizer to set the pH value close to neutral, stop the hydrolysis reaction, and initialize the polycondensation (gelation) reaction. Nitric acid (HNO_3_, assay 70%, Sigma Aldrich, France, LOT:STBJ3144) was used as a catalyst to partially hydrolyze TEOS and also for dissolving Eu_2_O_3_ and Dy_2_O_3._ Deionized water (DI H_2_O, ρ = 18.2 MΩ cm at 25 °C, total organic carbon up to 20 ppb, microorganisms < 10 CFU mL^−1^, heavy metals < 0.01 ppm, silicates < 0.01 ppm, and total dissolved solids < 0.03 ppm) was used as a solvent.

All the chemicals used in this work are reagent and analytical grade and were used without further purification.

### 2.2. Preparation of SrAl_2_O_4:_Eu, Dy, and B/SiO_2_ Composite Materials

In this study, Sr_0.97_Eu_0.01_Dy_0.02_Al_2-x_B_x_O_4_/SiO_2_ composite materials samples were synthesized by using the sol–gel method following the procedure described by Skuja et al. [22]. Additional steps were added to this synthesis process to obtain 2 wt% SrAl_2_O_4_:Eu, Dy, and B in SiO_2_ glass. According to Kajihara et al. [23,24,25], the formation of SrAl_2_O_4_:Eu, Dy, and B particles in the sol–gel-derived SiO_2_ glass matrix was expected to occur. Samples with different concentrations of boron (0 at% for comparison and three concentrations—7, 10, and 15 at% B—as the optimal concentrations for powder material by Vitola et al. [26] were synthesized and investigated, as it was previously shown to greatly affect the luminescent properties of the resulting sample [6]. The resulting crystallographic phase of the strontium aluminate can be different from SrAl_2_O_4_, however, the starting materials were chosen in proportions for 2 wt% SrAl_2_O_4_:Eu, Dy, and B, and the previous research suggests the creation of the simplest aluminate phase with a possible admixture of Sr_4_Al_14_O_25_ [26].

The first step of the sol–gel synthesis process consists of the slow addition of a dilute aqueous solution of 0.0181 mmol HNO_3_ (18.05 mmol L^−1^) to 25 mmol (5.58 mL) of TEOS. The resultant mixture was then stirred for 10 min at room temperature, and then the appropriate amounts of Sr(NO_3_)_2_ and Al(NO_3_)_3_·9H_2_O were added. Stirring was continued for 5 min, and, subsequently, the appropriate amounts of Eu^3+^, Dy^3+^, and B^+^ ion solutions were added dropwise to the reaction mixtures at room temperature under constant stirring speed. After the addition of each dopant ion solution, stirring was continued for 5 min to obtain a homogeneous reaction mixture, and then stirring was continued for an additional 50 min at room temperature. The clear, homogeneous reaction solutions were obtained with a pH value between 1 and 2. To increase the pH value to a range of 5–6, an aqueous solution of 3.378 mmol AcONH_4_ (73.92 mmol L^−1^) was slowly added to the reaction solutions at room temperature under constant stirring, and stirring was continued for 2 more minutes. The prepared sols, with a pH value around 5–6, were transferred into centrifuge tubes with a volume that did not exceed 60% of the volume of the centrifuge tubes. Then, these centrifuge tubes were sealed, and the gelation process was carried out at room temperature for 1 day. Subsequently, sealed centrifuge tubes containing the reaction gel were placed in a laboratory heating–drying oven, where gelation and formation of hydrogel processes were carried out at 50–60 °C for 2 days. After that, the hydrogels were aged at 80 °C for 2 weeks in sealed centrifuge tubes and then gently dried at 80 °C for 1 week in open centrifuge tubes. After the aging process, white, opaque SrAl_2_O_4_:Eu, Dy, and B/SiO_2_ composite xerogel samples were obtained. These xerogel samples were then sintered at 600 °C for 2 h with a heating rate of 5 °C min^−1^ in a reductive atmosphere (5% H_2_/95% Ar) to reduce Eu^3+^ into Eu^2+^ and eliminate nitric oxides and organic side-products. Then the samples were naturally cooled down to room temperature, and glassy, translucent samples were obtained (Figure 1).

The XRD method was used to confirm the composition of the sample. However, the spectrum does not allow for distinguishing the crystallographic phase of the strontium aluminate particles, as the particles are dispersed in the glass in a small concentration (Figure 1b). The spectra are similar to other borosilicate glass reports [27,28].

### 2.3. Methods of Characterization

Raman spectra were measured in backscattering configuration by an Andor Shamrock 303 spectrograph, Belfast, United Kingdom, with a 1200 L/mm grating and cooled silicon CCD, using 784.8 nm and 200 mW continuous wave (CW) excitation. The morphology of the samples was characterized by scanning electron microscopy (SEM, Tescan Lyra, Brno-Kohoutovice, Czech Republic) operated at 15 kV. Samples were characterized by X-ray diffraction (XRD, Rigaku MiniFlex 600 X-ray diffractometer, Neu-Isenburg, Germany) using a cathode voltage of 40 kV and a current of 15 mA with Cu Kα radiation (1.5418 Å). Photoluminescence (PL) and photostimulated (PSL) spectral measurements were conducted using a Horiba iHR320 monochromator, Oberursel, Germany, with a 150 L/mm diffraction grating. The monochromator was coupled with an Andor DV420A-BU2, Belfast, United Kingdom, CCD camera. Photoluminescence was excited with a CryLas Nd:YAG laser, Berlin, Germany (266 nm). The transmittance spectra were recorded with an Agilent—Cary 7000 spectrophotometer (Santa Clara, CA 95051, USA).

## 3. Results and Discussion

### 3.1. The Successes and Failures of PeL Silicate Glass Synthesis

In this study, we demonstrate the formation of SrAl_2_O_4_ doped with Eu, Dy, and B in a SiO_2_ glass structure using the one-pot low-temperature sol–gel synthesis method. By varying the synthesis conditions, we can use water-soluble precursors (e.g., nitrates) and a dilute aqueous solution of rare-earth (RE) nitrates as starting materials for SrAl_2_O_4_ synthesis, which can be formed during the sol–gel process at relatively low sintering temperatures (600 °C). The glass matrix is a suitable candidate for dispersing RE ions, co-doping with other oxides, or for a complex oxide material doped with RE ions because they can be integrated into both the amorphous and nanocrystalline phases [29,30,31].

The formation of SrAl_2_O_4_ phosphor in the SiO_2_ glass network (polymerization) takes place gradually by sintering the obtained xerogel at increasingly higher temperatures (600 °C), which is much lower than the usual temperatures used for solid-state synthesis of SrAl_2_O_4_ phosphor (1300 °C) and also for sol–gel synthesis (1200 °C) [15]. Cracking, which can occur during the drying and sintering processes, is the main issue that needs to be addressed during the synthesis process. According to the literature [4,5], a slow drying process is often used to prevent uneven gel shrinkage and reduce capillary pressure, which can lead to sample cracking. Therefore, we suggest that a slow drying procedure (2 weeks) is optimal for these samples to effectively remove excess water (mainly OH groups) and solvent molecules that can affect the SrAl_2_O_4_ phosphor’s luminescent properties. Initially, the gels were aged in closed centrifuge tubes for 1 week at 80 °C. Then, the tubes were opened, and the gels were gently dried for another week at 80 °C. We attempted to reduce the drying time to one week, but the resulting SrAl_2_O_4_/SiO_2_ glass composite material did not exhibit the characteristic luminescent properties or the long afterglow of Eu^2+^ in the SrAl_2_O_4_ matrix. Instead, it showed red Eu^3+^ luminescence without afterglow, which was likely due to residual water that hydrolyzed the SrAl_2_O_4_ phosphor, resulting in a non-luminescent aluminum hydroxide precipitate and soluble strontium hydroxide [18,32]. The samples that were created with the slow-drying process did not show luminescence degradation; this was tested 1 month after the initial sample synthesis.

Using HNO_3_ as a catalyst helps to obtain homogeneous, crack-free SrAl_2_O_4_/SiO_2_ glass composite samples. Additionally, during the sintering process at 600 °C, it is possible to remove any side products and organic compounds that may be left over after the drying process. In contrast, we attempted to use HF as a catalyst for sol–gel-derived SrAl_2_O_4_/SiO_2_ glass composite synthesis [30] to shorten the processing time and avoid fractures during the drying and sintering processes. HF is known to decrease gelation time and increase the average pore size of the resulting gels, thereby reducing the processing time. The presence of large pores is advantageous in preventing fractures and sample bloating during the drying and sintering processes due to the release of gaseous species such as CO_2_ and CO. However, the presence of large voids in the SrAl_2_O_4_/SiO_2_ glass composite material can affect its luminescence properties. Compared to standard gas-phase synthetic silica, these sol–gel-derived SrAl_2_O_4_/SiO_2_ glass composites contain a significant amount of nanopores in their structure (Figure 2). The internal surfaces of these nanopores contain free silanols (SiO–H), which can reduce the lifetime of the RE ion luminescence. It is also evident that the addition of boron affects the surface roughness and pore size of the resulting material. An excessive amount of boron (above 10%) creates a rougher surface with more pores.

Another issue that can influence the sufficient incorporation of SrAl_2_O_4_ and RE ions into the glass network is the use of solvents (e.g., alcohols), bases, drying control chemical additives (DCCA), or surfactants. It is crucial to understand the impact of these reagents because they can affect the final material composition, morphology, pore size distribution, etc. In addition, hydrolyzed groups in SiO_2_ play a significant role in the subsequent gel formation during sol–gel synthesis. Here, the molar ratio of TEOS to H_2_O was 1:22 (R = 22) to obtain a more homogeneous structure and to fully hydrolyze the ethyl groups in the silicon-organic precursor TEOS. Furthermore, the addition of Sr(NO_3_)_2_ and Al(NO_3_)_3_ precursors and a dilute aqueous solution of RE nitrate results in a homogeneous and clear solution with a pH of 2–3. Adding AcONH_4_ to increase the pH value (5–6) will cause gelation to occur more quickly and stop the hydrolysis reaction. Therefore, the molar ratio of TEOS to AcONH_4_ was 1:0.14 to increase the pH value closer to neutral.

Several publications have been devoted to doping RE ions into sol–gel-derived silica glass, which is commonly co-doped with P or Al. These two elements facilitate the dissolution of RE ions in silica glasses due to the selective coordination strength of PO_4_ or AlO_x_ units [23,24,25,33,34,35,36]. Another way to incorporate SrAl_2_O_4_ into SiO_2_ glass is to use ready-made SrAl_2_O_4_ phosphor powder obtained by solid-state or sol–gel synthesis. However, by choosing this kind of strategy, several steps are introduced to obtain SrAl_2_O_4_/SiO_2_ glass composite material, and as a result, the number of problems and the time spent obtaining materials increase. One of the important issues is that SrAl_2_O_4_ phosphor hydrolyzes in water into non-luminescent compounds. Therefore, it is crucial to improve the water stability of SrAl_2_O_4_ particles before adding them to the synthesis of sol–gel-derived SiO_2_ glass, such as by encapsulation with silica [37,38] Another important issue is that stable SrAl_2_O_4_ particles need to be created, which will disperse throughout the entire volume of the reaction solution and not sediment during the sol–gel synthesis and drying/aging processes. This property is strongly influenced by the morphology of the particles, and there is a high probability that the particles need to be stabilized with organic compounds, such as polymers or surfactants. Therefore, the most suitable synthesis of SrAl_2_O_4_ particles must be found, which will most likely be a multi-step synthesis. In contrast, we have demonstrated a one-pot additive-free sol–gel preparation method for SrAl_2_O_4_/SiO_2_ glass composites that does not require the use of organic solvents, DCCA, or surfactants. This approach not only reduces the number of steps, which can introduce errors and problems, but also reduces the time required to obtain materials and saves energy (e.g., no sintering above 1200 °C) and material resources (atom economy), making it a more environmentally friendly strategy.

### 3.2. The Optical Properties of the PeL Glass

Raman spectra of the samples show a glassy structure (Figure 3); it can be seen that small rings of silica tetrahedra, associated with the defect bands D1 and D2 at 480 and 600 cm^−1^ are present, similarly to [37].

For all SiO_2_ samples, a signal at ~970 cm^−1^ appears (see Figure 3), which is not normally present in pure SiO_2_ glass. In the frequency range from 800 to 1200 cm^−1^, bands near 960 cm^−1^ and 1115 cm^−1^ are observed for borate glasses (SrO–B_2_O_3_–SiO_2_ system) with relatively large Sr and B mol% to SiO_2_ [38]. In the present samples, the signal at ~970 cm^−1^ is slightly shifted to the right compared to the borate glasses signal (960 cm^−1^). Since SiO_2_ glass has a disordered structure, introducing the SrAl_2_O_4_: Eu, Dy, and B to porous SiO_2_ structure leads to the tendency by changing the Si–O–Si band length, local symmetry, and glass network during thermal treatment of sol–gel synthesized glasses. For such glasses, the chemical synthesis procedure and the drying and sintering parameters are very crucial since they can affect the final SiO_2_ glass structure [39].

Compared to “type III” wet synthetic silica glass, the present samples show a large concentration of three-membered Si–O rings (signal at ~600 cm^−1^, D2), with increasing boron (%B) concentration (7–15%B). The signal of four-membered Si–O rings (signal at ~480 cm^−1^, D1) is also observed. This indicates the possible formation of voids (pores) in the glass structure. The peak around ~800 cm^−1^ is related to the presence of characteristic Si–O–Si vibration bonds [39,40], and the peak around 980 cm^−1^ is a typical signal for porous SiO_2_ samples, on whose surfaces Si–OH groups are located [37]. However, a signal at 900–970 cm^−1^ is attributed to Si–O–NBO (non-bridging silicon–oxygen “NBO”) in sol–gel SiO_2_ glasses. Similarly to silicon, boron is a network former that forms the three-dimensional glass network and changes the glass structure through band shifting. B can form strong covalent bonds with anions (B–O–B bridges, B–O–Si, B–O–, BO_3_ triangles, BO_4_ tetrahedra, etc.) [1]. Thus, introducing B to the glass network leads to disruption of the continuous glass network due to the creation of local defects, for example, the formation of Si–O–NBO [41]. The signal of Si–O–NBO (at ~970 cm^−1^) can be used to study the possible incorporation of SrAl_2_O_4_:Eu, Dy with different B concentrations (%B) in the SiO_2_ structure.

Unfortunately, the Raman spectrum does not yield important information about the crystallographic phase of strontium aluminate, as the possible information is obscured by the glass Raman signal (i.e., for SrAl_2_O_4_, an intensive peak of 461 cm^−1^ and smaller intensity peaks at 782 and 1340 cm^−1^).

The emission spectra (Figure 4a) correspond to the typical 5d → 4f transition of Eu^2+^ and are broadband, with a maximum at 443 nm. The 580–710 nm region also exhibits Eu^3+^ emission lines [42], more prominently for samples with a 15% B concentration. It is observable that the maximum wavelength position is shifted to longer wavelengths with higher boron concentrations (10% and 15%), and these samples also decrease in afterglow time and intensity, as can be seen in Figure 2b. It is possible that the sample contains a mixture of different crystallographic phases of strontium aluminate, as for pure SrAl_2_O_4_, the emission wavelength is reported to be 520 nm [1], and therefore it can be speculated that maybe the sample contains mostly Sr_4_Al_14_O_25_. This, however, cannot be strongly concluded only by the emission spectra, as the emission strongly depends on the local environment of the luminescence center and could also be influenced by the incorporation of the glass. The emission during afterglow consists only of the Eu^2+^ broadband, as expected. The sample with no boron addition shows a very short afterglow. The time until the afterglow reaches 1% of the initial intensity does not exceed 20 s for any of the samples; however, this article outlines a new route of PeL glass synthesis, and more work on prolonging the afterglow intensity or tailoring the optical properties for a specific application is ahead.

The emission spectrum of the samples (Figure 4a) shows that boron is playing a crucial role in the formation of trapping centers needed for PeL processes. Boric acid acts as a flux and lowers melting temperature [6,26,43,44], and some publications conclude that it also creates substitutional BO_4_ units in the place of AlO_4_ tetrahedra, thus creating lattice distortions responsible for electron trapping [6,26], as borate (BO_4_) is more ionic in nature than AlO_4_ because of its smaller size and higher electronegativity of B compared to Al. It is visible that there is an optimal boron concentration for enhanced PeL properties of the glass, and above the optimal concentration, serious lattice deformation occurs—the emission peak is shifted to longer wavelengths (443 to 475 nm), the characteristic Eu^3+^ emission lines appear, indicating some amount of unreduced Eu^3+^ still remains in the sample, and the afterglow is diminished. This could indicate that with a larger amount of substitutional BO_4_ units, the lattice site for Sr (and, resultingly, Eu^2+^ or Eu^3+^) shrinks, therefore complicating the reduction. However, this is speculation and needs more experimental proof. We can support the claim that B addition presumably creates lattice distortions that can lead to electron trapping. Dy ions, when incorporated in the Sr lattice site, have an uncompensated charge, stimulating the charge trapping on BO_4_ units; however, a red shift of the emission occurs with larger boron concentrations. Some authors attribute it to the decrease in the average bond length between Eu^2+^ and O atoms and the reduction of cell volume that might happen when more AlO_4_ tetrahedra are substituted with BO_4_ units; therefore, enhancement of crystal field intensity and the red shift of the emission peak can occur [45].

The obtained samples are visibly translucent and without inhomogeneities observable with the naked eye. The samples show good transparency at visible and IR wavelengths (Figure 5). Transmittance spectra are similar for all samples; however, there are small differences in the shorter wavelength range; the local maxima in the range of 240–360 nm were more pronounced for the samples with higher boron concentrations (10 and 15%). It is known that the shape of the fundamental absorption edge in the exponential (Urbach) region is dependent on the disorder effects in the glass [46,47]. Therefore, we can conclude that excessive boron addition leads to a more disordered SiO_2_ network. The decrease of transmittance in 1200 and 1400 nm is attributed to vibrational overtones of C–H bonds by Pogareva et al. [48], and it is possible that some organic material is still present in the samples from the synthesis, as described in [22].

## 4. Conclusions

An economically promising method of persistently luminescent silicate glass synthesis that does not involve high temperatures or ready-made (separately synthesized) PeL particles is offered for the first time. The formation of PeL strontium aluminate doped with Eu, Dy, and B in a SiO_2_ (silicon dioxide) glass structure by low-temperature sol–gel synthesis (up to 600 °C) was demonstrated.

As a result, translucent, persistently luminescent glass is obtained. The glass shows the typical Eu^2+^ luminescence and the characteristic afterglow. Future work to prolong the afterglow is needed. It is concluded that the slow drying procedure (2 weeks) is optimal for these samples to sufficiently get rid of the excess water (mainly OH groups) and solvent molecules that can influence the strontium aluminate luminescence properties. It can also be concluded that boron plays a crucial role in the formation of trapping centers needed for PeL processes. Substitutional BO_4_ units emerge in the place of AlO_4_ tetrahedra, thus creating lattice distortions responsible for electron trapping; however, an excessive boron concentration leads to a rougher surface with more pronounced pores on the surface as well as a disorder in the SiO_2_ lattice, thus suppressing the luminescent properties of the material.

## Figures and Tables

**Figure 1 materials-16-04416-f001:**
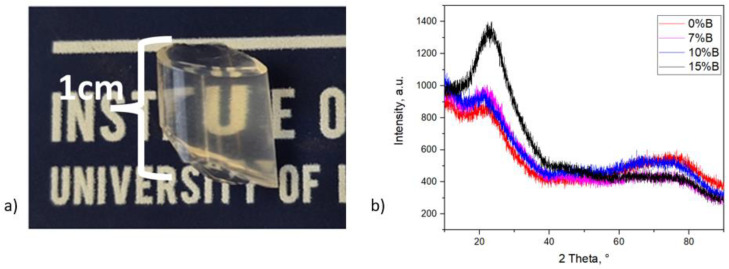
(**a**) Photo of the sample (with B 7%) after heat-treatment; (**b**) XRD of the samples.

**Figure 2 materials-16-04416-f002:**
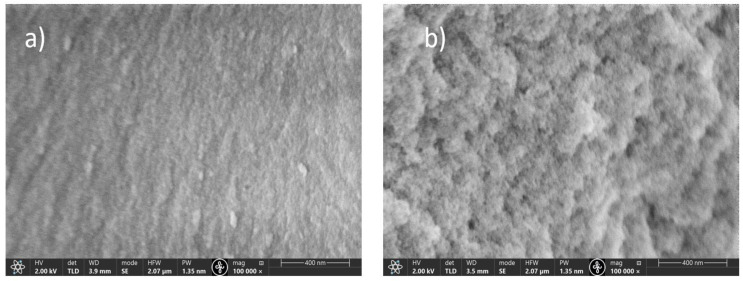
SEM images of samples with 7% (**a**) and 15% (**b**) boron concentrations.

**Figure 3 materials-16-04416-f003:**
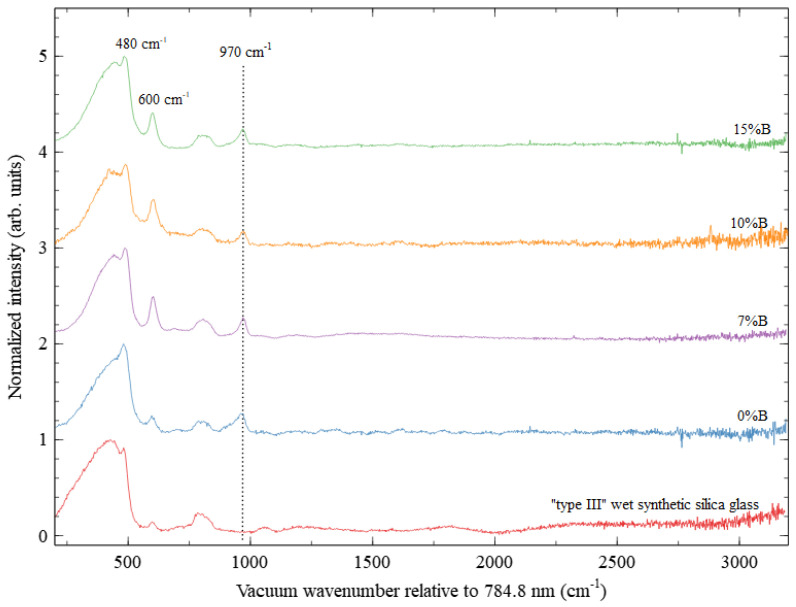
Raman spectra of SiO_2_@SrAl_2_O_4_: Eu, Dy, and B with different concentrations of boron (%B) and their comparison to the spectrum of commercial “type III” wet synthetic silica glass (bottom).

**Figure 4 materials-16-04416-f004:**
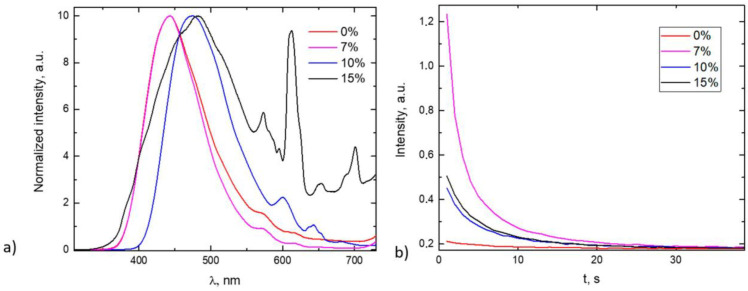
The emission spectra (**a**) and afterglow kinetics (**b**) for samples with different boron concentrations.

**Figure 5 materials-16-04416-f005:**
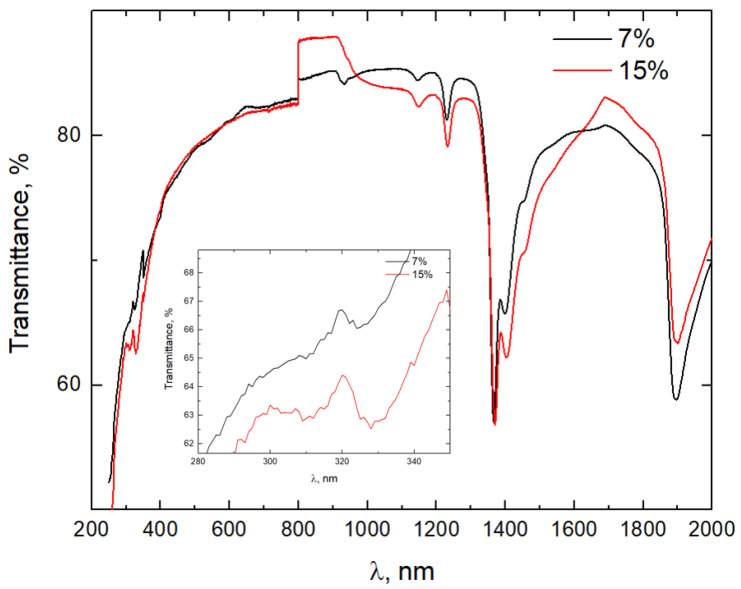
Transmittance spectra for the sample with 7% and 15% boron addition. Inset—240–380 nm range.

## Data Availability

The data of this research is readily available upon request.

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
