# Peer review of "Sol–Gel Synthesis of Translucent and Persistent Luminescent SiO2@ SrAl2O4 Eu, Dy, B Materials"

_materials, 2023, doi:10.3390/ma16124416_

Round 1

Reviewer 1 Report

Literature citation is missing at the line 198. Please correct it

Author Response

The citation has been added.

Author Response

Thank you for the detailed and useful comments! We appreciate your time and value your comments. I attach a detailed answer below.

Reviewer 3 Report

This is quite interesting research paper that probably can be recommended for publication, but only after clarifying and detailing some parts of the text.

1.     Introduction. 1 paragraph. Among the 7 supporting references, five refer to the authors. Does this mean that the authors are the main players in the field of persistent luminescent materials, or is it just excessive self-citation?

2.     2 paragraph.  There is no supporting references. Does this mean that these statements are being made for the first time?

3.     Line 40. Sentence “… containing Sr4Al14O25:Eu2+,Dy3+ microparticles conventionally in air [10,11].”  These references describe simpler materials: SrAl2O4 and CaAl2O4, but not mentioned in this sentence.

4.     The disadvantage of the introduction is the absolute absence of any information about the material of the study, namely SrAl2O4.

5.     It is important to note that these and similar spinel materials have many important applications as optical materials, or materials for nuclear/fusion technology, such as diagnostic/detector components and materials for shielding etc. 

See for example:

Luchechko, A., Zhydachevskyy, Y., Ubizskii, S. et al. Afterglow, TL and OSL properties of Mn2+-doped ZnGa2O4 phosphor. Sci Rep 9, 9544 (2019). https://doi.org/10.1038/s41598-019-45869-7

Pan, L., Wang, Y., Yin, L., Zhang, M., Li, Y., Townsend, P. D., & Poelman, D. Structural and optical properties of iron ions doped near-infrared persistent spinel-type phosphors. Journal of Luminescence, 258, 119822 (2023) https://doi.org/10.1016/j.jlumin.2023.119822

and references therein.

6.     In the abstract there is mention about (line 16) “excess water (mainly -OH groups)”. The next mention of this is only in Conclusions.

7.     It is known that there is already commercial SrAl2O4:Eu2+,Dy3+ MPs (Jinan G.L. New Materials, China, YG-101) (see info in reference [10]. In this regard, how justified is the synthesis and how comparable with commercial material is what is synthesized in this work?

8.     Line 70-71. Can you confirm the structure of the synthesized materials by X-ray diffraction and Raman. This is important because of quite high concentration of the dopants. How was stoichiometry checked?

9.     Where is boron located, what is its probability of being in the position of Sr and Al? The problem of antisite defects in MgAl2O4 is well studied

See, for example, Seeman, V., et al. "Fast-neutron-induced and as-grown structural defects in magnesium aluminate spinel crystals with different stoichiometry." Optical Materials 91 (2019) 42-49,  and references therein.

10.  Figure 1. It is necessary to add a scale in order to see the actual size of the sample.

11.  Is the sample free of point defects and how was this checked?

12.  Very often, in the case of high impurity concentrations, aging of the sample occurs and as a result, the impurity agglomeration, phase precipitation, etc. are observed. Has this been checked?

13.  Fig.3. how does the luminescence spectra change during afterglow?

14.  Fig.3. How and with what was this luminescence excited? If by photons, then what is the excitation spectrum for each of the luminescence subbands?

15.  Fig.3 (a). The presented spectra have not been analyzed and the interpretation of all the observed features has not been given. The latter would make it possible to more accurately understand the afterglow.

16.  For optical materials, Eg band gap data are absolutely important, but this information is missing for considered phosphor.

In general, the manuscript can be considering as interesting and can be recommended for publication after constructive reflection on the above comments.

Author Response

(The authors gave the same response as above.)

Round 2

Reviewer 2 Report

Thanks for the new report and addressing all the comments.

I have noticed that the references style is not uniform. Sometimes the complete list of authors is given, and sometimes the main author et al. formula is used.

Have the authors performed an XRD in the basic SrAl2O4, Eu, Dy samples doped with 7, 10 and 15% B? I mean, with regular powder samples not integrated into the glass? That could be a good idea to rule out the possible stabilization of the Sr4Al14O25 phase by boron. If it's the case, I would include it in the manuscript. 

Last, the color legend of the XRD presented is not correct for the highly B-doped samples.

Author Response

Thank you for your comments! 

We modified the legend and the colors in the XRD.

The references and et.al usage follows the guidelines - if there are more than 5 contributing authors it is converted to et.al.

For the powder material with varied boron - yes, we have checked the influence on the phase and it is visible, that more boron increases the Sr4Al14O25 phase, however it is still not the dominant phase. 

Vitola, V., Bite, I., Millers, D., Zolotarjovs, A., Laganovska, K., Smits, K., & Spustaka, A. (2020). The boron effect on low temperature luminescence of SrAl2O4:Eu, Dy. Ceramics International, 46(16), 26377–26381. doi:10.1016/j.ceramint.2020.01.208  10.1016/j.ceramint.2020.01.208    

Reviewer 3 Report

The authors have successfully improved the original version of their manuscript, responding constructively to all the comments/recommendations of the reviewer.  Therefore, the article can be recommended for publication.

Author Response

Thank you for your contribution!